# Smart phone-based transcutaneous electrical acupoint stimulation as adjunctive therapy for hypertension (STAT-H trial): protocol for a cluster randomised controlled trial

Jian-Feng Tu,[1] Si-Bo Kang,[1] Li-Qiong Wang,[1] Shi-Yan Yan,[1] Chao-Qun Yan [ID],[2] Xin-Tong Su,[1] Guang-Xia Shi [ID],[1] Bao-Hong Mi,[1] Ying Lin,[1] Yu Wang,[1] He-Wen Li,[1] Xue-Zhou Wang,[1] Xiao Wang,[1] Jing-Wen Yang [ID],[1] Cun-Zhi Liu [ID][1]

J-FT and S-BK contributed equally.

J-FT and S-BK are joint first authors.

For numbered affiliations see end of article.

**Correspondence to**
Professor Cun-Zhi Liu; lcz623780@126.com

## ABSTRACT

**Introduction** Hypertension is a common risk factor for cardiovascular disease. Transcutaneous electrical acupoint stimulation (TEAS) may be effective for hypertension, but the evidence remains limited. The aim of this study is to evaluate the effectiveness and safety of the smart phone-based TEAS as adjunctive therapy for hypertension.

**Methods and analysis** This study is a 52-week cluster randomised controlled trial with 1600 hypertension patients in 32 community health service centres. Patients who meet the inclusion criteria will be randomised into usual care group or TEAS group in a 1:1 ratio. All patients will be provided with usual care as recommended by the guidelines. In addition to this, patients in the TEAS group will receive non-invasive acupoint electrical stimulation for 30 min at home, 4 times weekly for 12 weeks. The primary outcome will be the mean difference in the changes in office systolic blood pressure from baseline to 12 weeks between TEAS and usual care groups. Secondary outcomes will include the change of mean diastolic blood pressure, proportion of patients with controlled blood pressure (blood pressure <140/90 mm Hg), proportion of patients taking antihypertensive drugs, change in number of antihypertensive drugs and changes in 12-item Short-Form. Tertiary outcomes will include change in body mass index, change in waist circumference, physical activity and medication adherence. Safety outcomes will be any serious adverse events and clinical events.

**Ethics and dissemination** This study has been approved by ethics committee of Beijing University of Chinese Medicine (No. 2020BZHYLL0104). Written informed consent will be obtained from all patients before randomisation. Trial results will be disseminated in peer-reviewed publications.

**Trial registration number** ChiCTR2000039400.

## STRENGTHS AND LIMITATIONS OF THIS STUDY

⇒ In this randomised controlled trial, a large population of 1600 hypertension patients will be recruited from 32 community health service centres.

⇒ This community-based cluster trial could lower the risk of contamination as much as possible and overcome the traditionally low detection of hypertension in hospital-based studies.

⇒ The smart phone-based transcutaneous electrical acupoint stimulation (TEAS) can be received at home by the patients themselves to reduce the burden of logistics and lost wages.

⇒ The background of TEAS software would recognise patients with poor compliance real timely and remind them of receiving TEAS to improve the adherence rate.

⇒ Surrogate outcome, not clinical event, is used as primary outcome and blinding for patients was not feasible in this study.

suggested that the global prevalence of hypertension was 20% for females and 25% for males.[2] Moreover, according to the latest American College of Cardiology/American Heart Association guideline,[3] the prevalence of hypertension increased to nearly 50%.[4–6] Hypertension accounted for 9.4 million deaths and 7.0% of global disability-adjusted life-years.[7] The total costs of hypertension will increase to an estimated US$274 billion by 2030.[8]

Lifestyle modifications are recommended by all guidelines in different countries,[3 9 10] while most patients are unable to sustain lifestyle changes. Antihypertensive drugs are widely prescribed for patients with hypertension. However, fewer than 20% of patients with hypertension have their blood pressure controlled.[10] The reasons for low control

## INTRODUCTION

Cardiovascular disease is the most common cause of premature death and disability worldwide, and hypertension is the most common risk factor for cardiovascular disease.[1] WHO

rate of blood pressure include the poor adherence of medicine and the burden of logistics. Identifying non-pharmaceutical therapy with good adherence that can be done at home is urgently needed to reduce blood pressure in patients with hypertension.

Acupuncture, as a non-pharmaceutical therapy, was used for several cardiovascular diseases.[11 12] Previous studies indicated that acupuncture could decrease blood pressure[13 14] and a recent systematic review found that acupuncture as adjunctive therapy may be effective for hypertension.[15] However, acupuncture also has the same limitation of logistical burdens as the antihypertensive drugs. Transcutaneous electrical acupoint stimulation (TEAS), a non-invasive acupuncture therapy, has a similar effect with acupuncture[16–18] and can be performed at home by the patients themselves. Our pilot study found that TEAS added to usual care for patients with hypertension was feasible and may be a potential treatment option.[19] The aim of this cluster randomised controlled trial is to evaluate the effectiveness and safety of the smart phone-based TEAS as adjunctive therapy for hypertension.

## METHODS
### Study design

This two-arm, open-label, cluster randomised controlled trial will be conducted at 32 community health service centres in China. The total trial period was 52 weeks, including 12 weeks of treatment and 40 weeks of follow-up. The study protocol (version number: 2.0; 13 June 2020) has been registered on Chinese clinical trial registry (No. ChiCTR2000039400). The protocol will be reported following the SPIRIT guideline.[20] Figure 1 shows the flow diagram of the trial.

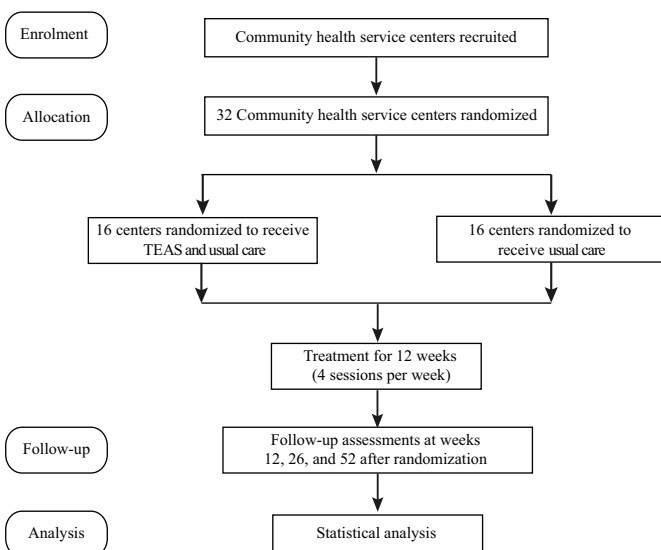

**Figure 1** Flow diagram. TEAS, transcutaneous electrical acupoint stimulation.

### Study recruitment

There are 16 districts and 346 community health service centres in Beijing, China. Of them, 32 community health service centres (online supplemental file 1), 2 centres in each district, will be recruited through Beijing Administration of Traditional Chinese Medicine and Beijing Association of Acupuncture-Moxibustion. Patients will be diagnosed as hypertension according to the European guideline[9] and Chinese guideline.[10] About 50 patients will be enrolled in each centre by their family doctors. The family doctors will conduct a preliminary screening to exclude ineligible patients, then the office blood pressure will be measured in the consulting room by clinical research coordinators to determine whether the patients will be finally included or not.

### Participants
#### Inclusion criteria

1. Office systolic blood pressure (SBP) from 140 to 159 mm Hg and/or diastolic blood pressure (DBP) from 90 to 99 mm Hg for patients who did not take antihypertensive drugs previously, or office SBP from 120 to 159 mm Hg and/or DBP from 80 to 99 mm Hg for patients who did not change antihypertensive drugs at least 1 month.
2. Aged between 18 and 65 years (male or female).
3. Answering and completing the questionnaires.
4. Using smart phone.
5. Written informed consent.

#### Exclusion criteria

1. Other diseases that can affect the blood pressure, including but not limited to diseases of renal parenchyma, renal artery stenosis, obstructive sleep apnoea syndrome, and primary hyperaldosteronism.
2. Receiving medicines that affect blood pressure except antihypertensive drugs in last month, including but not limited to glucocorticoids and central nervous system inhibitors.
3. Contraindications for the use of acupoint stimulators: wearing a pacemaker and other implanted medical devices; epilepsy; scars, bruises, scratches or inflammation on the skin of the acupoints; allergic to electrodes.
4. Uncontrolled diabetes (HbA1c≥6.5%).
5. Drug or alcohol abuse.
6. Pregnant or breast feeding.
7. Participated in other clinical trials in the past 3 months.

### Randomisation and masking

The unit of randomisation will be the community health service centres. Thirty-two centres will be randomised to either TEAS or usual care group in a 1:1 ratio with a central randomisation system. All patients within the defined centre will receive the same allocated treatment. Each centre will not be randomised until total 50 patients are recruited to avoid selection bias. The randomisation sequence will be generated by an independent statistician with the Stata V.12.0 software and will be stratified by

districts. The patients and the acupuncturists who teach patients how to locate acupoints and to receive TEAS at home will not be blinded. However, the family doctors who prescribe antihypertensive drugs, outcome assessors and statisticians will be blinded.

## Interventions

### Usual care group

All patients will be provided usual care at the respective community health service centre, in conjunction with any ongoing care that they are receiving from their family doctor. Patients at high or very high cardiovascular risk will receive antihypertensive drugs according to the recommendation of the guidelines,[9 10] including diuretics, β-blockers, calcium channel blockers, angiotensin receptor blockers and ACE inhibitors. The choice of antihypertensive drugs will be individualised, with consideration of patients' coexisting conditions and adverse effects to the medications. Research team will engage in consistent communication with each patient's family doctor, including a standard letter conveying the reminder that the doctor is free to implement any additional therapies that are deemed to be appropriate, including increase or decrease the type and dosage of antihypertensive drugs. Research team will also send education materials to all patients. Content will include information about hypertension, its clinical features, epidemiology, diagnostic criteria and treatment options, as well as the role and suggestions of lifestyle intervention in hypertension.

### TEAS group

In addition to the usual care, the patients in the TEAS group will receive 48 sessions of TEAS treatment (4 times weekly for 12 weeks). TEAS will be performed using a portable instrument (SDP-310, Hwato, Suzhou Medical Appliances, Suzhou, China) with a pair of rounded electrodes (diameter=38 mm) through smart phone. The adhesive electrodes will be applied to the skin as illustrated in figure 2 after the skin sites are cleaned with alcohol to avoid any barrier conduction of the electrical current. The instrument has ten levels of stimulus intensities, and patients asked to increase the stimulus intensity until local, rhythmic contractions of the muscle are obtained without producing pain. According to the previous studies,[16 19 21 22] bilateral *Hegu* (LI4), *Quchi* (LI11), *Zusanli* (ST36) and *Taichong* (LR3) will be used in our trial. On the first day, the electrodes will be placed at the ipsilateral LI4 and LI11, and the appropriate stimulus intensity will last for 15 min. Then the opposite arm will be stimulated in the same way for another 15 min. On the third day, the ST36 and LR3 will be stimulated for 15 min per leg in the same way. Two pairs of acupoints (LI4-LI11 and ST36-LR3) will be stimulated alternatively every other day. Locations of acupoints are shown in table 1. The acupuncturists will train the patients on how to locate the acupoints and use the instrument face to face. Instructions and acupoint maps will also be provided to help patients to locate the

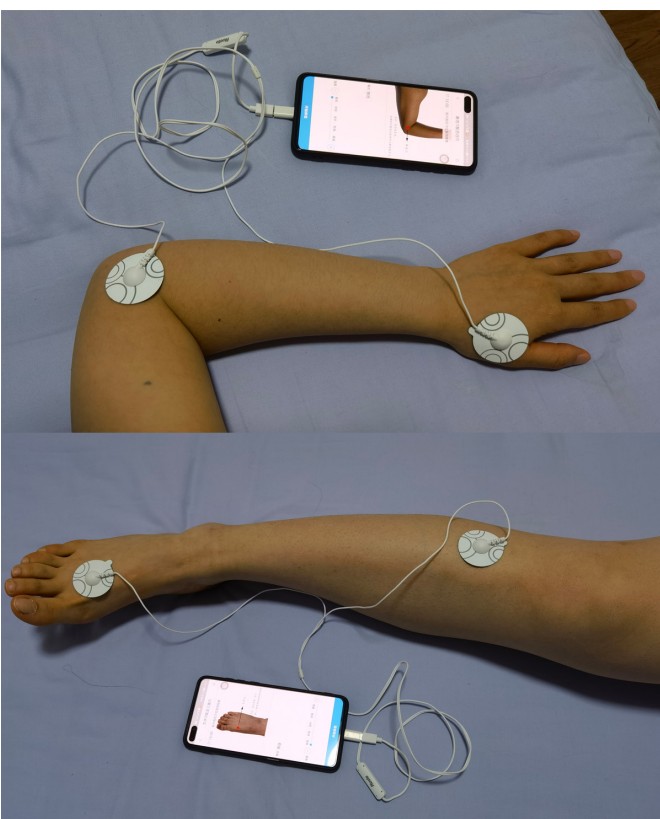

**Figure 2**  The instrument and the locations of acupoints.

acupoints at home. The patients will be told that the acupoints stimulation can be done when they are leisure at home, like reading the newspaper or watching television. The date, duration and specific acupoints of each acupoint stimuli will be recorded real timely by TEAS software. Thus, the investigator could recognise patients with poor compliance in a timely manner and prompt them to complete the predefined 48 sessions.

## Outcomes

The primary outcome will be the mean difference in the changes in office SBP from baseline to 12 weeks between TEAS and usual care groups. SBP will be measured as per the WHO STEPS protocol to ensure consistency of recordings, and an automated monitor (HEM-7136, Omron, Kyoto, Japan) will be used at each visit. Patients will be asked to rest quietly for at least 5 min in the sitting position before measuring blood pressure. Blood pressure of both arms will be measured at baseline with the upper arm at heart level, and the arm with higher blood pressure will be the one measured throughout the entire study. At each time point, the blood pressure measurement will be repeated 3 times every 5 min, and the last two will be averaged.[23 24]

Secondary outcomes will include differences in the changes in office DBP between the TEAS and usual care groups; the proportion of patients with well-controlled blood pressure (office blood pressure <140/90 mm Hg); the proportion of patients taking antihypertensive drugs;

**Table 1** Locations of acupoints in the TEAS group

| Acupoints | Locations |
| --- | --- |
| Taichong (LR3) | On the dorsum of the foot, between the first and second metatarsal bones, in the depression distal to the junction of the bases of the two bones, over the dorsalis pedis artery |
| Zusanli (ST36) | On the anterior aspect of the leg, on the line connecting ST35 with ST41, 3 cun* inferior to ST35 |
| Hegu (LI4) | On the dorsum of the hand, radial to the midpoint of the second metacarpal bone |
| Quchi (LI11) | On the lateral aspect of the elbow, at the midpoint of the line connecting LU5 with the lateral epicondyle of the humerus |

*One cun (≈20 mm) is defined as the width of the interphalangeal joint of patient's thumb.
TEAS, transcutaneous electrical acupoint stimulation.

differences in the changes in number of antihypertensive drugs; and differences in the changes in 12-item Short-Form.[25] Tertiary outcomes will include change in body mass index; change in waist circumference; physical activity assessed using International Physical Activity Questionnaire[26]; medication adherence assessed using the 8-item Morisky Medication Adherence Scale 8[27] in patients taking antihypertensive drugs in both groups; and TEAS adherence assessed using number of sessions completed in TEAS group. All secondary outcomes and tertiary outcomes will be measured at weeks 12, 26 and 52. Moreover, blinding test of outcome assessors will be measured at week 12.

Safety outcomes will be any serious adverse events and clinical events (death, myocardial infarction, stroke and cardiovascular-related hospitalisations) at weeks 12, 26 and 52 in both groups. The schedule of enrolment, intervention and assessments is shown in figure 3.

### Sample size calculation

The effective sample size (ESS) of 392 patients in each group is calculated for SBP on the basis of an alpha level of 0.05 (two sided), a beta level of 0.80, an expected between-group difference of 6 mm Hg,[19] an SD of 10 mm Hg and a superiority margin of 4 mm Hg,[28] if randomisation takes place at the individual level. With average 40 patients per community and an intracluster correlation of 0.01,[29] the ESS should be multiplied by a design effect of 1.39, leading to 560 patients in 14 communities per group (1120 patients in 28 communities in total). As our experience suggests that up to 10% of the randomised communities and up to 20% of the recruited patients may drop out of the study, we aim to recruit 32 communities (16 per group) including 50 patients each, resulting in a target sample size of 1600 recruited patients in total (800 per group).

### Statistical analysis

A full analysis set, a per-protocol set and a safety set will be used in this trial. The full analysis set will consist of all randomised patients according to the intention-to-treat principle. The full analysis set will be the primary analysis, and all analyses will be conducted for this population if not otherwise stated. The per-protocol set will include those who complete all follow-ups on time and without major violations. Major violations include but are not limited to not ineligibility according to the inclusion and exclusion criteria, receiving medicines that affect blood pressure except antihypertensive drugs, or not completing ≤38 sessions TEAS (80% of prespecified 48 sessions). The per-protocol set will be the secondary analysis set and will be used for the sensitivity analyses for the primary outcome. All those who receive at least one session of TEAS will be defined as the safety set, which is used for the safety analyses.

Baseline characteristics will be summarised by groups. Continuous variables will be described using the mean (SD), or the median (IQR) if the normality assumption is violated. Categorical variables will be described using the frequency (percentage). The significance level of superiority test will be set at one-sided 0.025 for the primary outcome. The significance level will be set at two-sided 0.05 for secondary outcomes. All analyses will be carried out using SAS V.9.3.

The change from baseline in office SBP at week 12 will be analysed by a mixed-effect model using baseline value as a covariate, treatment group as a fixed effect, and community and interaction between community and

|  | STUDY PERIOD | | | | |
| --- | --- | --- | --- | --- | --- |
|  | Enrolment | Allocation | Post-allocation | | Closeout |
| TIME POINT | Week -1 | Week 0 | Week 12 | Week 26 | Week 52 |
| ENROLMENT: | | | | | |
| [Eligibility screen] | × | | | | |
| [Informed consent] | × | | | | |
| [Randomization] | | × | | | |
| INTERVENTION: | | | | | |
| [TEAS+usual care] | | | ◆——————◆ | | |
| [usual care] | | | ◆——————◆ | | |
| ASSESSMENTS: | | | | | |
| [systolic blood pressure] | | × | × | × | × |
| [diastolic blood pressure] | | × | × | × | × |
| [blood pressure control rate] | | × | × | × | × |
| [proportion of patients taking antihypertensive drugs] | | × | × | × | × |
| [number of antihypertensive drugs] | | × | × | × | × |
| [12-Item Short-Form] | | × | × | × | × |
| [body mass index] | | × | × | × | × |
| [waist circumference] | | × | × | × | × |
| [physical activity] | | × | × | × | × |
| [medication adherence] | | × | × | × | × |
| [blinding test] | | | × | | |
| [serious adverse events] | | | × | × | × |
| [clinical events] | | | × | × | × |

**Figure 3** Schedule of enrolment, intervention and assessments. TEAS, transcutaneous electrical acupoint stimulation.

treatment as random effects. Multiple imputation will be used for missing data.[30] For the secondary and tertiary outcomes, continuous variables will be compared at all follow-up time points using a mixed-effect model with repeated measurement methods. If there is a normality violation in the continuous variables, a transformation will be performed before the test. Categorical variables will be compared using a generalised linear mixed-effect model. The consistency of treatment effects on the primary outcome will be explored in predefined subgroups, including sex, education and taking antihypertensive drugs or not and urban or rural community.

## Data management and quality control

Data will be collected by using a web-based Research Electronic Data Capture system (https://yu.life.sjtu.edu.cn/redcap/) developed by Shanghai Jiao Tong University. All researchers will receive training about data input, inquiry and modification. Independent data managers will check the system to ensure the integrity, accuracy and timeliness of data. Any modifications of the data can be traced in the Research Electronic Data Capture system. Moreover, patient information will be anonymous, including name, ID number and telephone number, to protect privacy. After the trial is completed, this system will be locked by the data management team. Then the researchers can no longer modify the data. An independent data and safety monitor board will be established to review and interpret the trial data every 6 months.

## Patient and public involvement

The first author (JFT) and other coresearchers in community health service centres are patients with hypertension. They were involved in the design and conduct of the trial. During the design stage, choice of outcome measures, the number of study visits and methods of recruitment were determined by discussions with patients. Once the trial has been published, a brief summary of the findings will be presented to attendees in the community health service centres.

## Ethics and dissemination

This study conforms to the principles of the Declaration of Helsinki. The study protocol (version number: 2.0; 13 June 2020) was approved by the ethics committee of Beijing University of Chinese Medicine (No. 2020BZHYLL0104). Meanwhile, written informed consent will be obtained by the clinical research coordinator from all patients before randomisation. Results of the trial are expected to be published in a peer-reviewed journal.

## DISCUSSION

To our knowledge, this is the first cluster randomised trial to investigate the effectiveness and safety of the smart phone-based TEAS as adjunctive therapy for hypertension.

This study is a community-based cluster randomised trial. There are several advantages for this design. First, contamination is one of the common biases, but it can be alleviated by the cluster design because only the communities randomly assigned to the TEAS group will have access to the instrument. Furthermore, communities were chosen with adequate geographical separation from each other. Second, patients will be recruited from their homes or communities. Unlike hospital-based studies, which only include those who seek medical help, our design will overcome the traditionally low detection of hypertension.

The TEAS in this trial can be received through smart phone at home by the patients themselves. The majority of hypertension patients had to travel by taxi or use public transportation to attend physician visits and prescribe antihypertensive drugs. If the effectiveness of TEAS is verified, this non-pharmaceutical therapy will result in additional savings of cost and burden. The time, expense and lost wages are also identified as barriers for controlling blood pressure. The smart phone-based TEAS at home may contribute to improving treatment adherence. Furthermore, several patients sometimes forget to take medicine, especially when they are busy working or tripping. As we mentioned before, we could use the background of TEAS software to recognise patients with poor compliance real timely and remind them to complete treatment.

The mechanism of acupuncture to reduce blood pressure is still unclear. It was found that TEAS could decrease sympathetic nervous system activity and increase parasympathetic nervous system activity,[31] which may contribute to reducing blood pressure. Moreover, TEAS was shown to have a vasodilator effect.[32] Our animal experiments also indicated that mitogen-activated protein kinases and β-adrenergic receptors were involved in the mechanism of acupuncture's amelioration of hypertension.[33 34]

Our study has several limitations. First, blinding for patients was not feasible; however, several steps will be implemented to minimise potential biases. Outcome assessors will be masked and an electronic data capture system with tablet has been developed to separate streams of data collection processes. Moreover, the blood pressure recordings were from automated devices to minimise biases for the key outcome. Second, surrogate outcomes are used in this study. There is a good evidence to support improved clinical outcomes with the reductions in blood pressure shown in other trials.[35 36] To show a benefit in clinical events, a much larger and longer trial would be required. Third, patients in the usual care group will be screened for cardiovascular disease risk at baseline, have their blood pressure assessed and be provided with information about hypertension education. After being aware of their elevated blood pressure, some patients in the usual care group might modify their behaviour and receive more treatment than clinical practice. This modification might reduce the real differences between the groups, which means that we may underestimate the effectiveness of TEAS.

This trial may provide high-quality evidence for the effectiveness of the smart phone-based TEAS as adjunctive therapy for hypertension. If the study hypothesis is confirmed, smart phone-based TEAS may help to attain the United Nations General Assembly Action Plan for a one-third reduction in premature mortality from cardiovascular disease.

## Trial status

Recruitment was started on 18 November 2020 and is expected to close in June 2022.

**Author affiliations**
[1]International Acupuncture and Moxibustion Innovation Institute, School of Acupuncture-Moxibustion and Tuina, Beijing University of Chinese Medicine, Beijing, China
[2]Department of Acupuncture and Moxibustion, Dongzhimen Hospital, Beijing University of Chinese Medicine, Beijing, China

**Acknowledgements** Appreciation to every participant in the trial and every personnel in recruitment sites for their contributions.

**Contributors** C-ZL conceived of the study. C-ZL, J-FT and L-QW initiated the study design. S-YY drew up the statistical plan. S-BK, C-QY, X-TS, G-XS, B-HM, YL, YW, H-WL, J-WY, X-ZW and XW helped with its implementation. J-FT and S-BK drafted the manuscript and all the authors critically revised the manuscript. C-ZL sought funding. C-ZL, J-FT and L-QW sought ethical approval. All authors contributed to the refinement of the study protocol and approved the final manuscript.

**Funding** This work was supported by National Key R&D Programme of China (grant number: 2019YFC1712100) and Project for Distinguished Young Scholars of Beijing University of Chinese Medicine (grant number: BUCM-2019-JCRC011).

**Disclaimer** The sponsors have no role in the design of the trial, collection, management, analysis, interpretation of data; writing of the report or the decision to submit the report for publication.

**Competing interests** None declared.

**Patient and public involvement** Patients and/or the public were involved in the design, or conduct, or reporting, or dissemination plans of this research. Refer to the Methods section for further details.

**Patient consent for publication** Consent obtained directly from patient(s)

**Provenance and peer review** Not commissioned; externally peer reviewed.

**ORCID iDs**
Chao-Qun Yan http://orcid.org/0000-0001-7921-3057
Guang-Xia Shi http://orcid.org/0000-0002-8834-2085
Jing-Wen Yang http://orcid.org/0000-0002-7031-3446
Cun-Zhi Liu http://orcid.org/0000-0001-8031-5667

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
