## [Reviewer comments · BMJ Open]

ARTICLE DETAILS

TITLE (PROVISIONAL)	Smart phone-based Transcutaneous electrical acupoint stimulation as Adjunctive Therapy for Hypertension (STAT-H trial): protocol for a cluster randomised controlled trial
AUTHORS	Tu, Jian Feng; Kang, Si-Bo; Wang, Li-Qiong; YAN, Shiyan; Yan, Chao-qun; Su, Xin-Tong; Shi, Guangxia; Mi, Bao-Hong; Lin, Ying; Wang, Yu; Li, He-Wen; Wang, Xue-Zhou; Wang, Xiao; Yang, Jing-Wen; Liu, Cun-Zhi

VERSION 1 – REVIEW

REVIEWER	Hae Jeong Nam Kyung Hee Univ
REVIEW RETURNED	23-Nov-2021

GENERAL COMMENTS	1. " Transcutaneous electrical acupoint stimulation (TEAS), a non-invasive acupuncture therapy, has similar effect with acupuncture " - Any reference? 2. I wonder why the treatment intensity was not defined equally to all patients. You say that the TENS has ten levels of stimulus intensities, and patients asked to increase the stimulus intensity until local, rhythmic contractions of the muscle are obtained without producing pain. I think that this kind of study with patients-self treatment, the method should be even more defined. I need an explanation for that. 3. Where is the instrument figure? Because there are many different types of TENS, most studies using TENS describe the tool in detail. Your research also needs a picture of how TEAS is being treated in detail. 4. It should also explain how to encourage participants over a long period of time and ensure that they do TEAS accurately.
--

REVIEWER	Prithwish Banerjee University Hospitals Coventry and Warwickshire NHS Trust, Department of Cardiology
REVIEW RETURNED	21-Jan-2022

GENERAL COMMENTS	This is an interesting study on transcutaneous electrical acupoint stimulation for hypertension. It is generally well presented apart from a few minor grammatical errors which can be corrected by reading through the paper again.. I am happy with the design and limitations.
---

VERSION 1 – AUTHOR RESPONSE

Reviewer: 1

Dr. Hae Jeong Nam, Kyung Hee Univ

Comments to the Author:

1. " Transcutaneous electrical acupoint stimulation (TEAS), a non-invasive acupuncture therapy, has similar effect with acupuncture " - Any reference?

Answers: Thank you for your valuable suggestions. To our knowledge, three small short-term trials¹⁻³ indicated that TEAS has the similar antihypertensive effect with acupuncture. We have added these three trials as references. (Page 62 of 83, line 15)

References

- [1] Jacobsson F, Himmelmann A, Bergbrant A, *et al.* The effect of transcutaneous electric nerve stimulation in patients with therapy-resistant hypertension. *J Hum Hypertens* 2000;14:795-8.
- [2] Williams T, Mueller K, Cornwall MW. Effect of acupuncture-point stimulation on diastolic blood pressure in hypertensive subjects: a preliminary study. *Phys Ther* 1991;71:523-9.
- [3] Kaada B, Flatheim E, Woie L. Low-frequency transcutaneous nerve stimulation in mild/moderate hypertension. *Clin Physiol* 1991;11:161-8.

2. I wonder why the treatment intensity was not defined equally to all patients. You say that the TENS has ten levels of stimulus intensities, and patients asked to increase the stimulus intensity until local, rhythmic contractions of the muscle are obtained without producing pain. I think that this kind of study with patients-self treatment, the method should be even more defined. I need an explanation for that.

Answers: Thank you for your advice. Individualized treatment is one of the characteristics of traditional Chinese medicine. Given the various tolerability of patient population for electrical stimulation, the treatment intensity was not defined equally to all patients. Moreover, the similar design of treatment intensity is adopted in other scholars' trials.^{1 2}

References

- [1] Kaada B, Flatheim E, Woie L. Low-frequency transcutaneous nerve stimulation in mild/moderate hypertension. *Clin Physiol* 1991;11:161-8.

[2] Silverdal J, Mourtzinis G, Stener-Victorin E, et al. Antihypertensive effect of low-frequency transcutaneous electrical nerve stimulation (TENS) in comparison with drug treatment. *Blood Press* 2012;21:306-10.

3. Where is the instrument figure? Because there are many different types of TENS, most studies using TENS describe the tool in detail. Your research also needs a picture of how TEAS is being treated in detail.

Answers: We are sorry for this neglect. We have added the instrument into Figure 2.

Figure 2. The instrument and the locations of acupoints

4. It should also explain how to encourage participants over a long period of time and ensure that they do TEAS accurately.

Answers: Thank you for your concern. We have added additional information about how to encourage participants over a long period of time and ensure that they do TEAS accurately. Firstly, the acupuncturists will train the patients on how to locate the acupoints and use the instrument face-to-face. Instructions and acupoint maps will also be provided to help patients to locate the acupoints at home. The patients will be told that the acupoints stimulation can be done when they are leisure at home, like reading the newspaper or watching TV. Secondly, the date, duration, and specific acupoints of each acupoint stimuli will be recorded real-timely by TEAS software. Thus, the investigator could recognise patients with poor compliance in a timely manner and prompt them to complete the predefined 48 sessions. (Page 66 of 83, line 41-59) Moreover, our pilot study showed that 86.7% participants attended 39 or more sessions (80% of targeted 48 sessions).¹

References

1. Tu JF, Wang LQ, Liu JH, *et al.* Home-based transcutaneous electrical acupoint stimulation for hypertension: a randomized controlled pilot trial. *Hypertens Res* 2021;44:1300-6.

Reviewer: 2

Prof. Prithwish Banerjee, University Hospitals Coventry and Warwickshire NHS Trust

Comments to the Author:

This is an interesting study on transcutaneous electrical acupoint stimulation for hypertension. It is generally well presented apart from a few minor grammatical errors which can be corrected by reading through the paper again.. I am happy with the design and limitations.

Answers: Thank you for your suggestions. We have reread and revised these grammatical errors with red words.